# MomentDiff: Generative Video Moment Retrieval from Random to Real

**Pandeng Li**[1]*, **Chen-Wei Xie**[2], **Hongtao Xie**[1]†, **Liming Zhao**[2],
**Lei Zhang**[1], **Yun Zheng**[2], **Deli Zhao**[2], **Yongdong Zhang**[1]

[1] University of Science and Technology of China, Hefei, China
[2] Alibaba Group

lpd@mail.ustc.edu.cn, {htxie, leizh23, zhyd73}@ustc.edu.cn
{eniac.xcw, lingchen.zlm, zhengyun.zy}@alibaba-inc.com
zhaodeli@gmail.com

## Abstract

Video moment retrieval pursues an efficient and generalized solution to identify the specific temporal segments within an untrimmed video that correspond to a given language description. To achieve this goal, we provide a generative diffusion-based framework called MomentDiff, which simulates a typical human retrieval process from random browsing to gradual localization. Specifically, we first diffuse the real span to random noise, and learn to denoise the random noise to the original span with the guidance of similarity between text and video. This allows the model to learn a mapping from arbitrary random locations to real moments, enabling the ability to locate segments from random initialization. Once trained, MomentDiff could sample random temporal segments as initial guesses and iteratively refine them to generate an accurate temporal boundary. Different from discriminative works (*e.g.,* based on learnable proposals or queries), MomentDiff with random initialized spans could resist the temporal location biases from datasets. To evaluate the influence of the temporal location biases, we propose two "anti-bias" datasets with location distribution shifts, named Charades-STA-Len and Charades-STA-Mom. The experimental results demonstrate that our efficient framework consistently outperforms state-of-the-art methods on three public benchmarks, and exhibits better generalization and robustness on the proposed anti-bias datasets. The code, model, and anti-bias evaluation datasets are available at `https://github.com/IMCCretrieval/MomentDiff`.

## 1 Introduction

Video understanding [1–9] is a crucial problem in machine learning [10–16], which covers various video analysis tasks, such as video classification and action detection. But both tasks above are limited to predicting predefined action categories. A more natural and elaborate video understanding process is the ability for machines to match human language descriptions to specific activity segments in a complex video. Hence, a series of studies [17–22] are conducted on Video Moment Retrieval (VMR), with the aim of identifying the moment boundaries ( *i.e.,* the start and end time) within a given video that best semantically correspond to the text query.

As shown in Fig. 1(a), early works address the VMR task by designing predefined dense video proposals ( *i.e.,* sliding windows [23–25], anchors [26] and 2D map [27] ). Then, the prediction segment is determined based on the maximum similarity score between dense proposals and the

---

*Interns at Alibaba Group

†Corresponding author

37th Conference on Neural Information Processing Systems (NeurIPS 2023).

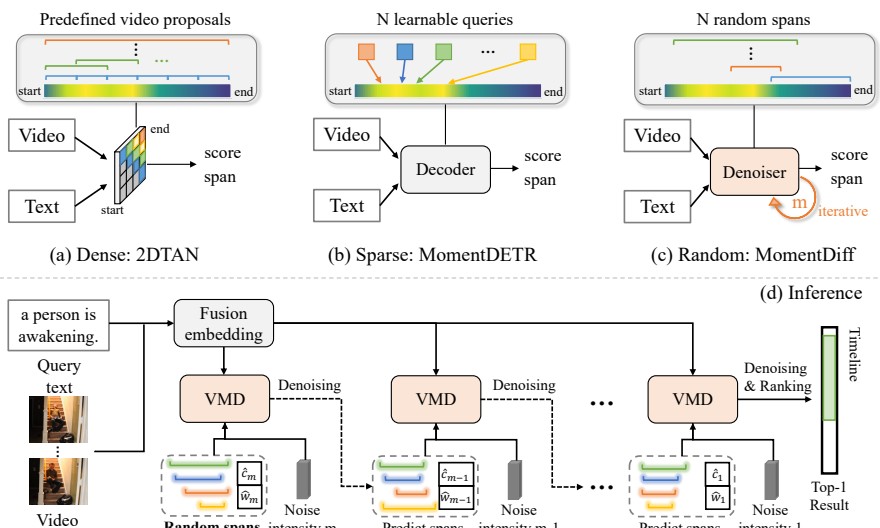

Figure 1: (a) Proposal-predefined methods. Yellow highlights in the timeline represent frequently occurring moments in the dataset. (b) Proposal-learnable methods. (c) Our generative method. (d) We model the VMR task as a process of gradually generating real temporal span from random noise.

query text. However, these methods have a large redundancy of proposals and the numbers of positive and negative proposals are unbalanced, which limits the learning efficiency [28]. To deal with this problem, a series of VMR works [29–36] have recently emerged, mainly discussing how to reduce the number of proposals and improve the quality of proposals. Among them, a promising scheme (Fig. 1(b)) is to use sparse and learnable proposals [34–36] or queries [37, 38] (*i.e.,* soft proposals) to model the statistics of the entire dataset and adaptively predict video segments. However, these proposal-learnable methods rely on a few specific proposals or queries to fit the location distribution of ground truth moments. For example, these proposals or queries may tend to focus on video segments where locations in the dataset occur more often (*i.e.,* yellow highlights in Fig. 1(b)). Thus, these methods potentially disregard significant events that transpire in out-of-sample situations. Recent studies [39, 40] indicate that VMR models [35, 27, 32] may exploit the location biases present in dataset annotations [23], while downplaying multimodal interaction content. This leads to the limited generalization of the model, especially in real-world scenarios with location distribution shifts.

To tackle the above issues, we propose a generative perspective for the VMR task. As shown in Fig. 1 (c) and (d), given an untrimmed video and the corresponding text query, we first introduce several random spans as the initial prediction, then employ a diffusion-based denoiser to iteratively refine the random spans by conditioning on similarity relations between the text query and video frames. A heuristic explanation of our method is that, it can be viewed as a way for humans to quickly retrieve moments of interest in a video. Specifically, given an unseen video, instead of watching the entire video from beginning to end (which is too slow), humans may first glance through random contents to identify a rough location, and finally iteratively focus on key semantic moments and generate temporal coordinates. In this way, we do not rely on distribution-specific proposals or queries (as mentioned in the above discriminative approaches) and exhibit more generalization and robustness (Tab. 5 and Fig. 4) when the ground truth location distributions of training and test sets are different.

To implement our idea, we introduce a generative diffusion-based framework, named MomentDiff. Firstly, MomentDiff extracts feature embeddings for both the input text query and video frames. Subsequently, these text and video embeddings are fed into a similarity-aware condition generator. This generator modulates the video embeddings with text embeddings to produce text-video fusion embeddings. The fusion embeddings contain rich semantic information about the similarity relations between the text query and each video frame, so we can use them as a guide to help us generate predictions. Finally, we develop a Video Moment Denoiser (VMD) that enhances noise perception and enables efficient generation with only a small number of random spans and flexible embedding learning. Specifically, VMD directly maps randomly initialized spans into the multimodal space,

taking them as input together with noise intensities. Then, VMD iteratively refines spans according to the similarity relations of fusion embeddings, thereby generating true spans from random to real.

Our main contributions are summarized as follows. 1) To the best of our knowledge, we are the first to tackle video moment retrieval from a generative perspective, which does not rely on predefined or learnable proposals and mitigates temporal location biases from datasets. 2) We propose a new framework, MomentDiff, which utilizes diffusion models to iteratively denoise random spans to the correct results. 3) We propose two "anti-bias" datasets with location distribution shifts to evaluate the influence of location biases, named Charades-STA-Len and Charades-STA-Mom. Extensive experiments demonstrate that MomentDiff is more efficient and transferable than state-of-the-art methods on three public datasets and two anti-bias datasets.

## 2 Related Work

**Video Moment Retrieval.** Video moment retrieval [32, 31, 41, 42, 33, 43–46] is a newly researched subject that emphasizes retrieving correlated moments in a video, given a natural language query. Pioneering works are proposal-based approaches, which employ a "proposal-rank" two-stage pipeline. Early methods [23, 25–27, 32] usually use handcrafted predefined proposals to retrieve moments. For example, CTRL [23] and MCN [25] aim to generate video proposals by using sliding windows of different scales. TGN [26] emphasizes temporal information and develops multi-scale candidate solutions through predefined anchors. 2DTAN [27] designs a 2D temporal map to enumerate proposals. However, these dense proposals introduce redundant computation with a large number of negative samples [28, 36]. Therefore, two types of methods are proposed: 1) Proposal-free methods [17, 31, 47] do not use any proposals and are developed to directly regress start and end boundary values or probabilities based on ground-truth segments. These methods are usually much faster than proposal-based methods. 2) Proposal-learnable methods that use proposal prediction networks [34–36] or learnable queries [37, 38] to model dataset statistics and adaptively predict video segments. QSPN [35] and APGN [34] adaptively obtain discriminative proposals without handcrafted design. LPNet [36] uses learnable proposals to alleviate the redundant calculations in dense proposals. MomentDETR [37] can predict multiple segments using learnable queries. Since proposal-learnable methods adopt a two-stage prediction [34–36] or implicit iterative [37] design, the performance is often better than that of proposal-free methods. However, proposal-learnable methods explicitly fit the location distribution of target moments. Thus, models are likely to be inclined to learn location bias in datasets [39, 40], resulting in limited generalization. We make no assumptions about the location and instead use random inputs to alleviate this problem.

**Diffusion models.** Diffusion Models [48–51] are inspired by stochastic diffusion processes in non-equilibrium thermodynamics. The model first defines a Markov chain of diffusion steps to slowly add random noise to the data, and then learns the reverse diffusion process to construct the desired data samples from the noise. The diffusion-based generation has achieved disruptive achievements in tasks such as vision generation [52–58] and text generation [59]. Motivated by their great success in generative tasks [60], diffusion models have been used in image perception tasks such as object detection [61] and image segmentation [62]. However, diffusion models are less explored for video-text perception tasks. This paper models similarity-aware multimodal information as coarse-grained cues, which can guide the diffusion model to generate the correct moment boundary from random noise in a gradual manner. Unlike DiffusionDet [61], we avoid a large number of Region of Interest (ROI) features and do not require additional post-processing techniques. To our knowledge, this is the first study to adapt the diffusion model for video moment retrieval.

## 3 Method

In this section, we first define the problem in Sec. 3.1, introduce our framework in Sec. 3.2, and describe the inference process in Sec. 3.3.

### 3.1 Problem Definition

Suppose an untrimmed video $\mathcal{V} = \{v_i\}_{i=1}^{N_v}$ is associated with a natural text description $\mathcal{T} = \{t_i\}_{i=1}^{N_t}$, where $N_v$ and $N_t$ represent the frame number and word number, respectively. Under this notation definition, Video Moment Retrieval (VMR) aims to learn a model $\mathbf{\Omega}$ to effectively predict the moment $\hat{x}_0 = (\hat{c}_0, \hat{w}_0)$ that is most relevant to the given text description: $\hat{x}_0 = \mathbf{\Omega}(\mathcal{T}, \mathcal{V})$, where $\hat{c}_0$ and $\hat{w}_0$ represent the center time and duration length of the temporal moments, *i.e.,* predicted spans.

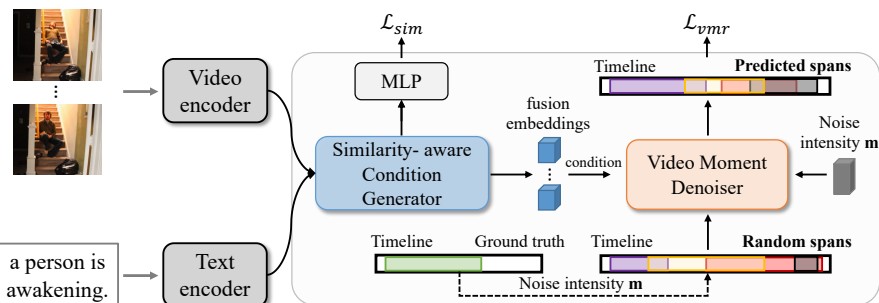

Figure 2: Our MomentDiff framework, which includes a Similarity-aware Condition Generator (SCG) and a Video Moment Denoiser (VMD). The diffusion process is conducted progressively in VMD.

## 3.2 The MomentDiff Framework

Fig. 2 sheds light on the generation modeling architecture of our proposed MomentDiff. Concretely, we first extract frame-level and word-level features by utilizing pre-trained video and text backbone networks. Afterward, we employ a similarity-aware condition generator to interact text and visual features into fusion embeddings. Finally, combined with the fusion embeddings, the video moment denoiser can progressively produce accurate temporal targets from random noise.

### 3.2.1 Visual and Textual Representations.

Before performing multimodal interaction, we should convert the raw data into a continuous feature space. To demonstrate the generality of our model, we use three distinct visual extractors [32, 44] to obtain video features $\mathcal{V}$: 1) 2D visual encoder, the VGG model [63]. 2) 3D visual encoder, the C3D model [64]. 3) Cross-modal pre-train encoder, the CLIP visual model [65]. However, due to the absence of temporal information in CLIP global features, we additionally employ the SlowFast model [66] to extract features, which concatenate CLIP features. Besides, to take full advantage of the video information [38], we try to incorporate audio features, which are extracted using a pre-trained PANN model [67]. To obtain text features, we try two feature extractors: the Glove model [68] and the CLIP textual model to extract 300-d and 512-d text features $\mathcal{T}$, respectively.

### 3.2.2 Similarity-aware Condition Generator

Unlike generation tasks [69] that focus on the veracity and diversity of results, the key to the VMR task is to fully understand the video and sentence information and to mine the similarities between text queries and video segments. To this end, we need to provide multimodal information to cue the denoising network to learn the implicit relationships in the multimodal space.

A natural idea is to interact and aggregate information between video and text sequences with a multilayer Transformer [70]. Specifically, we first use two multilayer perceptron (MLP) networks to map feature sequences into the common multimodal space: $\boldsymbol{V} \in \mathbb{R}^{N_v \times D}$ and $\boldsymbol{T} \in \mathbb{R}^{N_t \times D}$, where $D$ is the embedding dimension. Then, we employ two cross-attention layers to perform interactions between multiple modalities, where video embeddings $\boldsymbol{V}$ are projected as the query $\boldsymbol{Q}_v$, text embeddings $\boldsymbol{T}$ are projected as key $\boldsymbol{K}_t$ and value $\boldsymbol{V}_t$: $\hat{\boldsymbol{V}} = \text{softmax}\left(\boldsymbol{Q}_v \boldsymbol{K}_t^{\text{T}}\right) \boldsymbol{V}_t + \boldsymbol{Q}_v$, where $\hat{\boldsymbol{V}} \in \mathbb{R}^{N_v \times D}$. To help the model better understand the video sequence relations, we feed $\hat{\boldsymbol{V}}$ into a 2-layer self-attention network, and the final similarity-aware fusion embedding is $\boldsymbol{F} = \text{softmax}\left(\boldsymbol{Q}_{\hat{v}} \boldsymbol{K}_{\hat{v}}^{\text{T}}\right) \boldsymbol{V}_{\hat{v}} + \boldsymbol{Q}_{\hat{v}}$, where $\boldsymbol{Q}_{\hat{v}}, \boldsymbol{K}_{\hat{v}}, \boldsymbol{V}_{\hat{v}}$ is the matrix obtained from $\hat{\boldsymbol{V}}$ after three different projections respectively.

In the span generation process, even for the same video, the correct video segments corresponding to different text queries are very different. Since the fusion embedding $\boldsymbol{F}$ serves as the input condition of the denoiser, the quality of $\boldsymbol{F}$ directly affects the denoising process. To learn similarity relations for $\boldsymbol{F}$ in the multimodal space, we design the similarity loss $\mathcal{L}_{sim}$, which contains the pointwise cross entropy loss and the pairwise margin loss:

$$\mathcal{L}_{sim} = -\frac{1}{N_v} \sum_{i=1} \boldsymbol{y}_i * log(\boldsymbol{s}_i) + (1 - \boldsymbol{y}_i) * log(1 - \boldsymbol{s}_i) + \frac{1}{N_s} \sum_{j=1} \max\left(0, \beta + \boldsymbol{s}_{n_j} - \boldsymbol{s}_{p_j}\right), \quad (1)$$

where $s \in \mathbb{R}^{N_v}$ is the similarity score, which is obtained by predicting the fusion embedding $F$ through the MLP network. $y \in \mathbb{R}^{N_v}$ is the similarity label, where $y_i = 1$ if the $i$-th frame is within the ground truth temporal moment and $y_i = 0$ otherwise. $s_{p_j}$ and $s_{n_j}$ are the randomly sampled positive and negative frames. $N_s$ is the number of samples and the margin $\beta = 0.2$. Although $\mathcal{L}_{sim}$ may only help the fusion embedding retain some coarse-grained similarity semantics, this still provides indispensable multimodal information for the denoiser.

### 3.2.3 Video Moment Denoiser

Recent works [39, 40] have revealed that previous models [35, 27, 32] may rely on the presence of location bias in annotations to achieve seemingly good predictions. To alleviate this problem, instead of improving distribution-specific proposals or queries, we use random location spans to iteratively obtain real spans from a generative perspective. In this section, we first introduce the principle of the forward and reverse processes in diffusion models. Then, we build the diffusion generation process in the video moment denoiser with model distribution $p_\theta(x_0)$ to learn the data distribution $q(x_0)$.

**Forward process.** During training, we first construct a forward process that corrupts real segment spans $x_0 \sim q(x_0)$ to noisy data $x_m$, where $m$ is the noisy intensity. Specifically, the Gaussian noise process of any two consecutive intensities [49] can be defined as: $q(x_m \mid x_{m-1}) = \mathcal{N}(x_m; \sqrt{1-\beta_m}x_{m-1}, \beta_m I)$, where $\beta$ is the variance schedule. In this way, $x_m$ can be constructed by $x_0$: $q(x_{1:m} \mid x_0) = \prod_{i=1}^{m} q(x_i \mid x_{i-1})$. Benefiting from the reparameterization technique, the final forward process is simplified to:

$$x_m = \sqrt{\bar{\alpha}_m}x_0 + \sqrt{1-\bar{\alpha}_m}\epsilon_m, \qquad (2)$$

where the noise $\epsilon_m \sim \mathcal{N}(0, I)$ and $\bar{\alpha}_m = \prod_{i=1}^{m}(1-\beta_i)$.

**Reverse process.** The denoising process is learning to remove noise asymptotically from $x_m$ to $x_0$, and its traditional single-step process can be defined as:

Figure 3: Video moment denoiser. For simplicity, we only draw the intensity-aware attention structure that is different from the general Transformer.

$$p_\theta(x_{m-1} \mid x_m) = \mathcal{N}(x_{m-1}; \mu_\theta(x_m, m), \sigma_m^2 I) \qquad (3)$$

where $\sigma_m^2$ is associated with $\beta_m$ and $\mu_\theta(x_m, m)$ is the predicted mean. In this paper, we train the Video Moment Denoiser (VMD) to reverse this process. The difference is that we predict spans from the VMD network $f_\theta(x_m, m, F)$ instead of $\mu_\theta(x_m, m)$.

**Denoiser network.** As shown in Fig. 3, the VMD network mainly consists of 2-layer cross-attention Transformer layers. Next, we walk through how VMD works step by step. For clarity, the input span and output prediction presented below are a single vector.

❶ **Span normalization.** Unlike generation tasks, our ground-truth temporal span $x_0$ is defined by two parameters $c_0$ and $w_0$ that have been normalized to $[0, 1]$, where $c_0$ and $w_0$ are the center and length of the span $x_0$. Therefore, in the above forward process, we need to extend its scale to $[-\lambda, \lambda]$ to stay close to the Gaussian distribution [71, 61]. After the noise addition is completed, we need to clamp $x_m$ to $[-\lambda, \lambda]$ and then transform the range to $[0, 1]$: $x_m = (x_m/\lambda + 1)/2$, where $\lambda = 2$.

❷ **Span embedding.** To model the data distribution in multimodal space, we directly project the discrete span to the embedding space through the Fully Connected (FC) layer: $x'_m = \text{FC}(x_m) \in \mathbb{R}^D$. Compared to constructing ROI features in DiffusionDet [61], linear projection is very flexible and decoupled from conditional information (*i.e.,* fusion embeddings), avoiding more redundancy.

❸ **Intensity-aware attention.** The denoiser needs to understand the added noise intensity $m$ during denoising, so we design the intensity-aware attention to perceive the intensity magnitude explicitly. In Fig. 3, we use sinusoidal mapping for the noise intensity $m$ to obtain $e_m \in \mathbb{R}^D$ in the multimodal space and add it to the span embedding. We project $x'_m + e_m$ as query embedding and the positional embedding $pos_m \in \mathbb{R}^D$ is obtained by sinusoidal mapping of $x_m$. We can obtain the input query: $Q_m = \text{Concat}(\text{Proj}(x'_m + e_m), pos_m)$. Similarly, The input key is $K_f = \text{Concat}(\text{Proj}(F), pos_f)$ and the input value is $V_f = \text{Proj}(F)$, where $\text{Proj}(\cdot)$ is the projection function and $pos_f \in \mathbb{R}^D$ is the standard position embedding in Transformer [70]. Thus, the intensity-aware attention is:

$$Q_m = \text{softmax}\left(Q_m K_f^\text{T}\right)V_f + Q_m. \qquad (4)$$

❹ **Denoising training.** Finally, the generated transformer output is transformed into predicted spans $\hat{\boldsymbol{x}}_{m-1} = (\hat{\boldsymbol{c}}_{m-1}, \hat{\boldsymbol{w}}_{m-1})$ and confidence scores $\hat{\boldsymbol{z}}_{m-1}$, which are implemented through a simple FC layer, respectively. Following [71], the network prediction should be as close to ground truth $\boldsymbol{x}_0$ as possible. In addition, inspired by [37, 72], we define the denoising loss as:

$$\mathcal{L}_{\text{vmr}}\left(\boldsymbol{x}_0, f_\theta(\boldsymbol{x}_m, m, \boldsymbol{F})\right) = \lambda_{\text{L1}} \|\boldsymbol{x}_0 - \hat{\boldsymbol{x}}_{m-1}\| + \lambda_{\text{iou}}\, \mathcal{L}_{\text{iou}}\left(\boldsymbol{x}_0, \hat{\boldsymbol{x}}_{m-1}\right) + \lambda_{\text{ce}} \mathcal{L}_{ce}(\hat{\boldsymbol{z}}_{m-1}), \quad (5)$$

where $\lambda_{\text{L1}}$, $\lambda_{\text{iou}}$ and $\lambda_{\text{ce}}$ are hyperparameters, $\mathcal{L}_{\text{iou}}$ is a generalized IoU loss [73], $\mathcal{L}_{\text{ce}}$ is a cross-entropy loss. Note that the above procedure is a simplification of training. Considering that there may be more than one ground truth span in the dataset [37], we set the number of input and output spans to $N_r$. For the input, apart from the ground truth, the extra spans are padded with random noise. For the output, we calculate the matching cost of each predicted span and ground truth according to $\mathcal{L}_{vmr}$ (*i.e.,* the Hungarian match [72]), and find the span with the smallest cost to calculate the loss. In $\mathcal{L}_{\text{ce}}$, we set the confidence label to 1 for the best predicted span and 0 for the remaining spans.

### 3.3 Inference

After training, MomentDiff can be applied to generate temporal moments for video-text pairs including unseen pairs during training. Specifically, we randomly sample noise $\hat{\boldsymbol{x}}_m$ from a Gaussian distribution $\mathcal{N}(0, \mathbf{I})$, the model can remove noise according to the update rule of diffusion models [69]:

$$\hat{\boldsymbol{x}}_{m-1} = \sqrt{\bar{\alpha}_{m-1}} f_\theta(\hat{\boldsymbol{x}}_m, m, \boldsymbol{F}) + \sqrt{1 - \bar{\alpha}_{m-1} - \sigma_m^2} \frac{\hat{\boldsymbol{x}}_m - \sqrt{\bar{\alpha}_m} f_\theta(\hat{\boldsymbol{x}}_m, m, \boldsymbol{F})}{\sqrt{1 - \bar{\alpha}_m}} + \sigma_m \boldsymbol{\epsilon}_m. \quad (6)$$

As shown in Fig 1(d), we iterate this process continuously to obtain $\hat{\boldsymbol{x}}_0$ from coarse to fine. Note that in the last step we directly use $f_\theta(\hat{\boldsymbol{x}}_1, 1, \boldsymbol{F})$ as $\hat{\boldsymbol{x}}_0$. In $\hat{\boldsymbol{x}}_0$, we choose the span with the highest confidence score in $\hat{\boldsymbol{z}}_0$ as the final prediction. To reduce inference overhead, we do not employ any post-processing techniques, such as box renewal in DiffusionDet [61] and self-condition [71].

## 4 Experiments

### 4.1 Datasets, Metrics and Implementation Details

**Datasets.** We evaluate the efficacy of our model by conducting experiments on three representative datasets: Charades-STA [23], QVHighlights [37] and TACoS [74]. The reason is that the above three datasets exhibit diversity. Charades-STA comprises intricate daily human activities. QVHighlights contains a broad spectrum of themes, ranging from everyday activities and travel in lifestyle vlogs to social and political events in news videos. TACoS mainly presents long-form videos featuring culinary activities. The training and testing divisions are consistent with existing methods [28, 38].

**Metrics.** To make fair comparisons, we adopt the same evaluation metrics as those used in previous works [38, 37, 23, 75, 29, 19, 27], namely R1@n, MAP@n, and $\text{MAP}_{avg}$. Specifically, R1@n is defined as the percentage of testing queries that have at least one correct retrieved moment (with an intersection over union (IoU) greater than n) within the top-1 results. Similarly, MAP@n is defined as the mean average precision with an IoU greater than n, while $\text{MAP}_{avg}$ is determined as the average MAP@n across multiple IoU thresholds [0.5: 0.05: 0.95].

**Implementation details.** For a fair comparison [44, 20, 30], we freeze the video encoder and text encoder and use only the extracted features. For VGG [63], C3D [64] or SlowFast+CLIP (SF+C) [66, 65], we extract video features every 1/6s, 1s or 2s. So the frame number $N_v$ is related to the length of the video, while the max text length $N_t$ is set to 32. Note that since the videos in TACoS are long, we uniformly sample the video frame features and set the max frame number $N_v$ to 100 in TACoS. We set the hidden size $D = 256$ in all Transformer layers. In SCG, we also use a variant of the pairwise margin loss called InfoNCE loss [76]. The number of random spans $N_r$ is set to 10 for QVHighlights, 5 for Charades-STA and TACoS. We use the cosine schedule for $\beta$. For all datasets, we optimize MomentDiff for 100 epochs on one NVIDIA Tesla A100 GPU, employ Adam optimizer [77] with 1e-4 weight decay and fix the batch size as 32. The learning rate is set to 1e-4. By default, the loss hyperparameters $\lambda_{\text{L1}} = 10$, $\lambda_{\text{iou}} = 1$ and $\lambda_{\text{ce}} = 4$. The weight values for $\mathcal{L}_{sim}$ and $\mathcal{L}_{vmr}$ are 4 and 1. To speed up the sampling process during inference, we follow DDIM [69] and iterate 50 times.

Table 1: Performance comparisons (%) on the Charades-STA dataset. "⋆" denotes that we re-implement the method under the same training scheme. "A" stands for using audio data.

| Method | Type | Charades-STA | | | | |
|---|---|---|---|---|---|---|
| | | R1@0.5 | R1@0.7 | MAP@0.5 | MAP@0.75 | MAP$_{avg}$ |
| MAN [33] | VGG, Glove | 41.21 | 20.54 | - | - | - |
| RaNet⋆[75] | VGG, Glove | 42.91 | 25.82 | 53.28 | 24.41 | 28.55 |
| 2DTAN⋆[27] | VGG, Glove | 41.34 | 23.91 | 54.68 | 24.15 | 29.26 |
| DORi [78] | VGG, Glove | 43.47 | 26.37 | - | - | - |
| CBLN [44] | VGG, Glove | 47.94 | 28.22 | - | - | - |
| DCM [40] | VGG, Glove | 47.80 | 28.00 | - | - | - |
| MMN⋆ [28] | VGG, Glove | 46.93 | 27.07 | 58.85 | 28.16 | 31.58 |
| MomentDETR⋆ [37] | VGG, Glove | 50.54 | 28.01 | 57.39 | 25.62 | 29.87 |
| MomentDiff | VGG, Glove | **51.94** | **28.25** | **59.86** | **29.11** | **31.66** |
| UMT [38] | VGG+A, Glove | 48.44 | 29.76 | 58.03 | 27.46 | 30.37 |
| MomentDiff | VGG+A, Glove | **52.62** | **29.93** | **60.69** | **29.74** | **31.81** |
| DEBUG[79] | C3D, Glove | 37.39 | 17.69 | - | - | - |
| LPNet[36] | C3D, Glove | 40.94 | 21.13 | - | - | - |
| VSLNet⋆[30] | C3D, Glove | 48.67 | 30.33 | 56.88 | 25.79 | 30.16 |
| MomentDETR⋆[37] | C3D, Glove | 50.49 | 29.95 | 56.27 | 26.08 | 29.92 |
| MomentDiff | C3D, Glove | **53.79** | **30.18** | **59.32** | **29.85** | **31.89** |
| MomentDETR⋆ [37] | SF+C, C | 53.22 | 30.87 | 58.86 | 26.43 | 30.43 |
| MomentDiff | SF+C, Glove | 55.42 | 32.17 | 60.93 | 32.47 | 32.59 |
| MomentDiff | SF+C, C | **55.57** | **32.42** | **61.07** | **32.51** | **32.85** |

## 4.2 Performance Comparisons

**Comparison with state-of-the-art methods.** To prove the effectiveness of MomentDiff, we compare the retrieval performance with 17 discriminative VMR methods. Tab. 1, Tab. 2, and Tab. 3 show the R1@n, MAP@n, and MAP$_{avg}$ results on Charades-STA, QVHighlights and TACoS. Compared with SOTA methods [28, 38, 75, 44, 30, 37], MomentDiff achieves significant improvements on Charades-STA regardless of whether 2D features (VGG), multimodal features (VGG+A), 3D features (C3D), or multimodal pre-trained features (SF+C) are used. This proves that MomentDiff is a universal generative VMR method. In the other two datasets (QVHighlights and TACoS), we still have highly competitive results. Specifically, compared to MomentDETR [37], MomentDiff obtains 2.35%, 3.86%, and 13.13% average gains in R1@0.5 on three datasets. It is worth noting that TACoS contains long videos of cooking events where different events are only slightly different in terms of cookware, food and other items. The learnable queries in MomentDETR may not cope well with such fine-grained dynamic changes. We attribute the great advantage of MomentDiff over these methods to fully exploiting similarity-aware condition information and progressive refinement denoising.

Table 2: Performance comparisons (%) on QVHighlights Table 3: Performance comparisons (%) with SF+C video features and CLIP text features. "⋆" de-on TACoS. We adopt C3D features to notes that we re-implement the method with only segment encode videos. MDE is the abbreviation moment labels. "†" stands for using audio data. MDE is the of MomentDETR [37]. abbreviation of MomentDETR [37].

| Method | QVHighlights | | | | |
|---|---|---|---|---|---|
| | R1@0.5 | R1@0.7 | MAP@0.5 | MAP@0.75 | MAP$_{avg}$ |
| MCN [25] | 11.41 | 2.72 | 24.94 | 8.22 | 10.67 |
| CAL [80] | 25.49 | 11.54 | 23.40 | 7.65 | 9.89 |
| XML [81] | 41.83 | 30.35 | 44.63 | 31.73 | 32.14 |
| XML+ [81] | 46.69 | 33.46 | 47.89 | 34.67 | 34.90 |
| MDE⋆ [37] | 53.56 | 34.09 | 53.97 | 28.65 | 29.39 |
| MomentDiff | **57.42** | **39.66** | **54.02** | **35.73** | **35.95** |
| UMT⋆† [38] | 56.26 | 40.31 | 52.77 | 36.82 | 35.79 |
| MomentDiff† | **58.21** | **41.48** | **54.57** | **37.21** | **36.84** |

| Method | TACoS | | |
|---|---|---|---|
| | R1@0.1 | R1@0.3 | R1@0.5 |
| CTRL [23] | 24.32 | 18.32 | 13.30 |
| SCDM [32] | - | 26.11 | 21.17 |
| DRN [31] | - | - | 23.17 |
| DCL [41] | 49.36 | 38.84 | 29.07 |
| CBLN [44] | 49.16 | 38.98 | 27.65 |
| FVMR [42] | 53.12 | 41.48 | 29.12 |
| RaNet [75] | - | 43.34 | 33.54 |
| MDE⋆ [37] | 41.16 | 32.21 | 20.55 |
| MMN⋆ [28] | 51.39 | 39.24 | 26.17 |
| MomentDiff | **56.81** | **44.78** | **33.68** |

**Transfer experiments.** To explore the location bias problem, we first organize the Out of Distribution (OOD) experiment following [82], which repartitions Charades-STA [23] and ActivityNet-Captions [83] datasets according to moment annotation density values [82]. In Tab. 4, we exceed

Table 4: Transfer experiments (%) on Charades-CD (VGG, Glove) and ActivityNet-CD (C3D, Glove).

| Method | Charades-CD | | | | ActivityNet-CD | | | |
|---|---|---|---|---|---|---|---|---|
| | R1@0.3 | R1@0.5 | R1@0.7 | MAP$_{avg}$ | R1@0.3 | R1@0.5 | R1@0.7 | MAP$_{avg}$ |
| 2DTAN [27] | 49.71 | 28.95 | 12.78 | 12.60 | 40.04 | 22.07 | 10.29 | 12.77 |
| MMN [28] | 55.91 | 34.56 | 15.84 | 15.73 | 44.13 | 24.69 | 12.22 | 15.06 |
| MomentDETR [37] | 57.34 | 41.18 | 19.31 | 18.95 | 39.98 | 21.30 | 10.58 | 12.19 |
| MomentDiff | **67.73** | **47.17** | **22.98** | **22.76** | **45.54** | **26.96** | **13.69** | **16.38** |

Table 5: Transfer experiments (%) on anti-bias datasets with VGG and Glove video-text features.

| Method | Charades-STA-Len | | | | Charades-STA-Mom | | | |
|---|---|---|---|---|---|---|---|---|
| | R1@0.3 | R1@0.5 | R1@0.7 | MAP$_{avg}$ | R1@0.3 | R1@0.5 | R1@0.7 | MAP$_{avg}$ |
| 2DTAN [27] | 39.68 | 28.68 | 17.72 | 22.79 | 27.81 | 20.44 | 10.84 | 17.23 |
| MMN [28] | 43.58 | 34.31 | 19.94 | 26.85 | 33.58 | 27.20 | 14.12 | 19.18 |
| MomentDETR [37] | 42.73 | 34.39 | 16.12 | 24.02 | 29.94 | 22.16 | 11.56 | 18.66 |
| MomentDiff | **51.25** | **38.32** | **23.38** | **28.19** | **48.39** | **33.59** | **15.71** | **21.37** |

MomentDETR by a large margin on Charades-CD and ActivityNet-CD. These results prove the robustness of the model in dealing with OOD scenarios.

To further explore the impact of a single factor ($w_0$ or $c_0$) on the location bias problem, we organize moment retrieval on two anti-bias datasets with location distribution shifts: ❶ Charades-STA-Len. We collect all video-text pairs with $w_0 \leq 10s$ and randomly sample pairs with $w_0 > 10s$ in the original training set of Charades-STA, which account for 80% and 20% of the new training set, respectively. On the contrary, we collect all pairs with $w_0 > 10s$ and randomly sample pairs with $w_0 \leq 10s$ from the original test set, accounting for 80% and 20% of the new test set. ❷ Charades-STA-Mom. Similarly, we collect all video-text pairs with the end time $c_0 + w_0/2 \leq 15s$ and sample pairs with the start time $c_0 - w_0/2 > 15s$ as the training set, which accounts for 80% and 20%, respectively. Likewise, the construction rules for the test set are the opposite of those for the training set. Dataset statistics can refer to Train/Test in Fig. 4 and the supplementary material.

In Tab. 5, the proposed MomentDiff shows much more robustness than previous state-of-the-art method MomentDETR [37]. Concretely, compared with the experiment in Tab. 1, the performance gap between MomentDiff and MomentDETR gets larger on Charades-STA-Len and Charades-STA-Mom. Fig. 4 also demonstrates that the distribution of our prediction is closer to the one of the test set. We conjecture that it is because MomentDiff discards the learnable proposals that fit the prior distribution of the training set. Moreover, 2DTAN [27] and MMN [28] perform worse than MomentDETR on the original Charades-STA dataset, but they achieve better or comparable results than MomentDETR in Tab. 5. This shows that predefined proposals [27, 28] may be better than learnable proposals in dealing with the location bias problem, but they take up more time and space overhead. Differently, our method performs well on both public datasets and anti-bias datasets.

## 4.3 Ablation Study

To provide further insight into MomentDiff, we conduct critical ablation studies on Charades-STA.

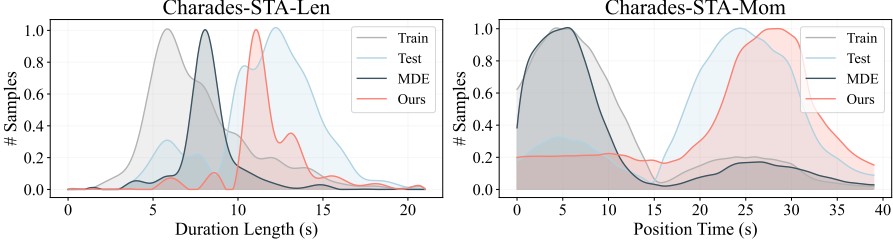

Figure 4: Statistical distributions on anti-bias datasets with location distribution shifts. We count the length or moment of each video-text pair, and finally normalize the counts (# Samples) for clarity. Train/Test: the count distribution of true spans on the training or test set. MDE: the count distribution of top-1 predicted spans by using MomentDETR on the test set.

Table 6: Ablation study (%) on the Charades-STA dataset with SF+C video features and CLIP text features. We report R1@0.5, R1@0.7 and $\text{MAP}_{avg}$. Default settings are marked in blue.

(a) Different span embedding types.

| Type | R1@0.5 | R1@0.7 | $\text{MAP}_{avg}$ |
|---|---|---|---|
| ROI | 50.21 | 28.85 | 29.12 |
| FC | **55.57** | **32.42** | **32.85** |

(b) Effect of scale $\lambda$.

| scale | R1@0.5 | R1@0.7 | $\text{MAP}_{avg}$ |
|---|---|---|---|
| 1.0 | 54.33 | 32.17 | 31.39 |
| 2.0 | **55.57** | **32.42** | **32.85** |
| 3.0 | 50.12 | 28.08 | 30.47 |

(c) VMD and noise intensity $m$.

| Type | R1@0.5 | R1@0.7 | $\text{MAP}_{avg}$ |
|---|---|---|---|
| w/o VMD | 46.69 | 24.03 | 23.28 |
| w/o $m$ | 50.41 | 29.73 | 28.72 |
| Ours | **55.57** | **32.42** | **32.85** |

(d) Different loss designs.

| $\mathcal{L}_{sim}$ | $\mathcal{L}_{vmr}$ | R1@0.5 | R1@0.7 | $\text{MAP}_{avg}$ |
|---|---|---|---|---|
| ✓ | | 31.53 | 16.52 | 17.89 |
| | ✓ | 40.94 | 24.65 | 23.76 |
| ✓ | ✓ | **55.57** | **32.42** | **32.85** |

(e) Effect of span number $N_r$.

| $N_r$ | R1@0.5 | R1@0.7 | $\text{MAP}_{avg}$ |
|---|---|---|---|
| 1 | 50.83 | 26.17 | 26.98 |
| 5 | **55.57** | **32.42** | **32.85** |
| 10 | 53.89 | 30.48 | 30.27 |
| 20 | 50.42 | 27.62 | 27.53 |

(f) Model performance vs. speed.

| Step | R1@0.5 | R1@0.7 | $\text{MAP}_{avg}$ | FPS |
|---|---|---|---|---|
| 1 | 53.31 | 29.78 | 29.54 | 531.4 |
| 2 | **55.62** | 31.92 | 32.99 | 465.0 |
| 10 | 55.41 | 32.16 | **33.54** | 338.2 |
| 50 | 55.57 | **32.42** | 32.85 | 148.8 |
| 100 | 55.39 | 32.39 | 32.93 | 97.89 |

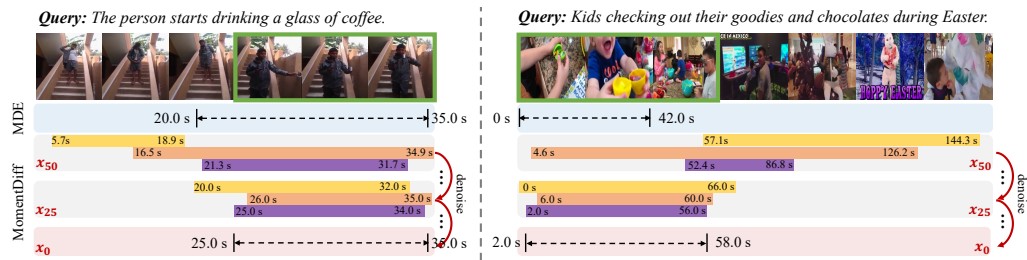

Figure 5: Visualization of the diffusion process on Charades-STA (Left) and QVHighlights (Right). For clarity, we show 3 random spans ($\boldsymbol{x}_{50}$) with Gaussian initialization, and progressively get the top-1 result ($\boldsymbol{x}_0$) according to the confidence score. Green box: right segment. MDE: MomentDETR [37].

**Span embedding type.** Regarding the way discrete spans are mapped to the embedding space, we compare the ROI strategy [61] with our linear projection (FC) in Tab. 6(a). For the ROI strategy, we slice the fusion embeddings $\boldsymbol{F}$ corresponding to random spans, followed by mean pooling on the sliced features. Tab. 6(a) shows that ROI does not work well. This may be due to two points: 1) ROI is a hard projection strategy, while the importance of each video frame is quite different. FC is similar to soft ROI, and its process can be trained end-to-end. 2) FC is decoupled from $\boldsymbol{F}$, which allows the model to focus on modeling the diffusion process and avoid over-dependence on $\boldsymbol{F}$.

**Scale $\lambda$.** $\lambda$ is the signal-to-noise ratio [61] of the diffusion process, and its effect is shown in Tab. 6(b). We find that the effect of larger $\lambda$ drops significantly, which may be due to the lack of more hard samples for denoising training when the proportion of noise is small, resulting in poor generalization.

**Video Moment Denoiser (VMD) and noise intensity $m$.** In Tab. 6(c), we first remove the denoiser and the diffusion process (w/o VMD). After training with the same losses, we find that predicting with only fusion embeddings $\boldsymbol{F}$ leads to a drastic drop in results, which reveals the effectiveness of denoising training. Then we remove the noise intensity $m$ (w/o $m$), and the result is reduced by 5.16% on R1@0.5. This shows that explicitly aggregating noise intensity with random spans improves noise modeling. Combined with VMD and $m$, the diffusion mechanism can fully understand the data distribution and generate the real span from coarse to fine.

**Loss designs.** In Tab. 6(d), we show the impact of loss functions. In $\mathcal{L}_{sim}$, we use pointwise and pairwise constraints to guide token-wise interactions between multimodal features, while ensuring reliable conditions for subsequent denoising. In $\mathcal{L}_{vmr}$, the model can learn to accurately localize exact segments. Adequate multimodal interaction and denoising training procedures are complementary.

**Span number.** In Tab. 6(e), we only need 5 random spans to achieve good results. Unlike object detection [61], the number of correct video segments corresponding to text query is small. Therefore, a large number of random inputs may make the model difficult to train and deteriorate the performance.

**Model performance vs. speed.** In Tab. 6(f), we explore the effects of different diffusion steps. When step=2, good results and fast inference speed have been achieved. Subsequent iterations can improve the results of high IoU (*i.e.,* R1@0.7), which shows the natural advantages of diffusion models.

### 4.4 Qualitative Results

We show two examples of the diffusion process in Fig. 5. We can find that the retrieved moments by MomentDiff are closer to the ground truth than those by MomentDETR. The diffusion process can gradually reveal the similarity between text and video frames, thus achieving better results. Besides, the final predictions corresponding to spans with multiple random initial locations are close to the ground truth. This shows that our model achieves a mapping from arbitrary locations to real segments.

## 5 Limitation and Conclusion

**Limitation.** Compared to existing methods [27, 37], the diffusion process requires multiple rounds of iterations, which may affect the inference speed. As shown in Tab. 6(f), we reduce the number of iterations, with only a small sacrifice in performance. In practical usage, we suggest choosing a reasonable step number for a better trade-off between performance and speed.

**Conclusion.** This paper proposes a novel generative video moment retrieval framework, MomentDiff, which simulates a typical human retrieval style via diffusion models. Benefiting from the denoising diffusion process from random noise to temporal span, we achieve the refinement of prediction results and alleviate the location bias problem existing in discriminative methods. MomentDiff demonstrates efficiency and generalization on multiple diverse and anti-bias datasets. We aim to stimulate further research on video moment retrieval by addressing the inadequacies in the framework design, and firmly believe that this work provides fundamental insights into the multimodal domain.

## 6 Acknowledgement

This work is supported by the National Key Research and Development Program of China (2022YFB3104700), the National Nature Science Foundation of China (62121002, 62022076, 62232006).

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

This supplementary material provides more details of our MomentDiff framework:

1. Implementation details.

2. Inference efficiency of MomentDiff.

3. More experiment results.

4. Broader impacts.

## A  Implementation details

### A.1  Datasets

**Public datasets. Charades-STA** [23] serves as a benchmark dataset for the video moment retrieval task and is built upon the Charades dataset, originally collected for video action recognition and video captioning. The Charades-STA dataset comprises 6,672 videos and 16,128 video-query pairs, allocated for training (12,408 pairs) and testing (3,720 pairs). On average, the videos in this dataset have a duration of 29.76 seconds. Each video is annotated with an average of 2.4 moments, with each moment lasting approximately 8.2 seconds. **QVHighlights** [37] contains 10,148 videos, each 150 seconds long and annotated with at least one text query describing its relevant content. These videos are from three main categories, daily vlogs, travel vlogs, and news events. On average, there are approximately 1.8 non-overlapping moments per query, annotated on 2s non-overlapping clips. The dataset contains a total of 10,310 queries with 18,367 annotated moments. The training set, validation set and test set include 7,218, 1,550 and 1,542 video-text pairs, respectively. **TACoS** [74] is compiled specifically for video moment retrieval and dense video captioning tasks. It is comprised of 127 videos that depict cooking activities, with an average duration of 4.79 minutes. TACoS contains a total of 18,818 video-query pairs. In comparison to the Charades-STA dataset, TACoS has more video segments that are temporally annotated with queries per video. On average, each video contains 148 queries. Additionally, the TACoS dataset is known for its difficulty, as the queries it contains are limited to only a few seconds or even just a few frames. To ensure impartial comparisons, we use the same dataset split [31], which consists of 10,146, 4,589, and 4,083 video-query pairs for the training, validation, and testing sets, respectively. **ActivityNet-Captions** [83] comprises of 20,000 videos and 100,000 descriptions that encompass a wide range of contexts. Similar to [27], we designate val 1 as the validation set and val 2 as the testing set. The dataset contains 37,417, 17,505, and 17,031 pairs of moments and corresponding sentences for training, validation, and testing respectively.

Table 7: Training and test sets on two anti-bias datasets.

| Dataset | Charades-STA-Len | | | Charades-STA-Mom | | |
|---|---|---|---|---|---|---|
| | $w_0 \leq 10s$ | $w_0 > 10s$ | Total | $c_0 + w_0/2 \leq 15s$ | $c_0 - w_0/2 > 15s$ | Total |
| Training | 9307 | 2326 | 11633 | 5330 | 1332 | 6662 |
| Test | 197 | 788 | 985 | 259 | 1038 | 1297 |

**Anti-bias datasets.** To explore the location bias problem, we construct two anti-bias datasets with location distribution shifts based on the Charades-STA dataset. In video moment retrieval, length and position are important parameters of spans.

Therefore, we first investigate the effect of span length $w_0$. As shown in Tab. 7, in the training set of Charades-STA-Len, we collect 9,307 video-text pairs with span length $w_0 \leq 10s$ and 2,326 video-text pairs with $w_0 > 10s$, accounting for 80% and 20% of the total training set. In contrast, in the test set, we select 197 video-text pairs with $w_0 \leq 10s$ and 788 video-text pairs with $w_0 > 10s$, accounting for 20% and 80% of the total test set.

Then, we design the dataset Charades-STA-Mom based on the span's end time $c_0 + w_0/2$ and start time $c_0 - w_0/2$. In the training set of Charades-STA-Mom, we collect 5,330 video-text pairs with $c_0 + w_0/2 \leq 15s$ and 1,332 video-text pairs with $c_0 - w_0/2 > 15s$, accounting for 80% and 20% of the total training set. In contrast, in the test set, we select 259 video-text pairs with $c_0 + w_0/2 \leq 15s$ and 788 video-text pairs with $c_0 - w_0/2 > 15s$, accounting for 20% and 80% of the total test set.

**Algorithm 1:** MomentDiff Training in a PyTorch-like style.

```
# Video features:  v_feats ∈ ℝ^{N_v×D}
# Text features:  t_feats ∈ ℝ^{N_t×D}
# Ground truth spans:  gt_spans : [∗, 2]
# alpha_cumprod(m):  cumulative product of α_i
# Fully-connected layer:  FC()
# Video Moment Denoiser:  VMD()
def train(v_feats, t_feats, gt_spans):
    # Similarity-aware Condition Generator
    f_feats = SCG(v_feats, t_feats)  #  N_v × D

    # Span normalization
    ps = pad_spans(gt_spans)  # Pad gt_spans to [N_r, 2]
    ps = (ps * 2 -1) * λ  # Signal scaling
    m = randint(0, M)  # Noise intensity
    noi = normal(mean=0, std=1) # Noise
    ps_m = sqrt( alpha_cumprod(m)) * ps +
            sqrt(1 - alpha_cumprod(m)) * noi  # Noisy span
    ps_m = (ps_m/λ +1)/2  # Normalization

    # Span embedding
    ps_emb_m = FC(ps_m)

    # Intensity-aware attention
    output_m = VMD(ps_emb_m, m, f_feats)  # Output embedding

    # Denoising training
    hat_ps_m = Span_pred(output_m)  # Predicted span
    hat_cs_m = Score_pred(output_m)  # Confidence score
    # Computing loss
    loss = loss_sim(f_feats) + loss_vmr(hat_ps_m, hat_cs_m, gt_spans)

    return loss
```

## A.2 Pseudo Code of MomentDiff

Algorithm 1 provides the pseudo-code of MomentDiff Training in a PyTorch-like style.

The inference procedure of MomentDiff is a denoising sampling process from noise to temporal spans. Starting from spans sampled in Gaussian distribution, the model progressively refines its predictions, as shown in Algorithm 2.

# B  Inference Efficiency of MomentDiff

**Inference time.** Inference efficiency is critical for machine learning models. We test 2DTAN [27], MMN [28], MomentDETR [37] and MomentDiff on the Pytorch framework [84] in Tab. 8. We test all models with one NVIDIA Tesla A100 GPU.

Compared with 2DTAN [27] and MMN [28], MomentDiff (Step=1) not only achieves the best results on recall, but also improves the inference speed by 5-7 times. This is because 2DTAN and MMN predefine a large number of proposals, which may be redundant and increase computational overhead. Compared to MomentDETR [37], MomentDiff (Step=10) achieves better results with similar inference time. The possible reasons are that we adopt fewer random spans, very simple network structures, and avoid post-processing.

**Algorithm 2:** MomentDiff inference in a PyTorch-like style.

```
# Video features:  v_feats ∈ ℝ^{N_v×D}
# Text features:  t_feats ∈ ℝ^{N_t×D}
# Video Moment Denoiser:  VMD()
def test(v_feats, t_feats, sampling_num):
    # Similarity-aware Condition Generator
    f_feats = SCG(v_feats, t_feats)  #  N_v × D

    # Noisy span:[N_r, 2]
    ps_m = normal(mean=0, std=1)

    # uniform sample
    intensity = reversed(linspace(-1, M, sampling_num))
    # [(M-1, M-2), (M-2, M-3), ..., (1, 0), (0, -1)]
    intensity_pairs = list(zip(intensity[:-1], intensity[1:]))

    for intensity_now, intensity_next in zip(intensity_pairs):
        # predict ps_0 from ps_m
        output_m = VMD(ps_m, f_feats, intensity_now)
        # Predicted span
        hat_ps_m = Span_pred(output_m)
        # Update ps_m
        ps_m = ddim_update(hat_ps_m, ps_m, intensity_now, intensity_next)

    # Confidence score
    hat_cs_m = Score_pred(output_m)

    return hat_ps_m, hat_cs_m
```

Table 8: The inference time of 2DTAN [27], MMN [28], MomentDETR [37] and MomentDiff on Charades-STA with VGG video features and Glove text features. We report R1@0.5, R1@0.7 and $MAP_{avg}$. Default settings are marked in  blue .

| Method | Charades-STA | | | Inference time |
| --- | --- | --- | --- | --- |
| | R1@0.5 | R1@0.7 | $MAP_{avg}$ | (second) |
| 2DTAN [27] | 41.34 | 23.91 | 29.26 | 42.18 |
| MMN [28] | 46.93 | 27.07 | 31.58 | 53.42 |
| MomentDETR [37] | 50.54 | 28.01 | 29.87 | 12.42 |
| MomentDiff (Step=1) | 49.17 | 26.39 | 29.12 | 7.56 |
| MomentDiff (Step=2) | 50.81 | 27.84 | 31.27 | 8.23 |
| MomentDiff (Step=10) | **52.36** | 28.08 | **31.75** | 11.01 |
| MomentDiff (Step=50) | 51.94 | 28.25 | 31.66 | 20.74 |
| MomentDiff (Step=100) | 52.21 | **28.84** | 31.01 | 34.35 |

# C   More Experiment Results

## C.1   Experiment on ActivityNet-Captions

The results of our method on ActivityNet-Captions are shown in Tab. 9. We record the 1 epoch training time and the inference time for all test samples in one NVIDIA Tesla A100 GPU. Our model has been improved compared with baseline (MomentDETR). Compared to SOTAs, we still have competitive results. Compared with MMN, our method is 25 times faster in training time and 7.24 times faster in testing time.

Table 9: Performance comparisons (%) on ActivityNet-Captions with C3D video features and Glove text features.

| Method | R1@0.3 | R1@0.5 | R1@0.7 | MAP$_{avg}$ | Training time | Inference time |
|---|---|---|---|---|---|---|
| 2DTAN [27] | 59.92 | 44.63 | 27.53 | 27.26 | 1.1h | 523.74s |
| MMN [28] | **65.21** | **48.26** | **28.95** | **28.74** | 1.5h | 662.12s |
| MomentDETR [37] | 61.87 | 43.19 | 25.74 | 25.63 | **0.05h** | **52.72s** |
| MomentDiff | 62.79 | 46.52 | 28.43 | 28.19 | 0.06h | 91.43s |

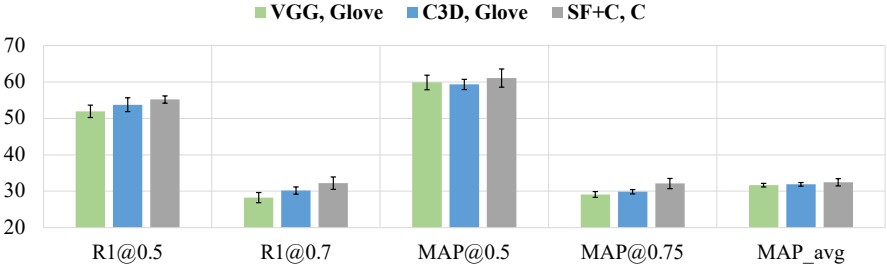

Figure 6: Performance fluctuations (%) corresponding to different features and multiple random seeds on the Charades-STA dataset.

## C.2  Error bars

Fig. 6 shows the performance fluctuation of the model on the Charades-STA dataset. We use different random seeds (seed= 2023, 2022, 2021, 2020, 2019) and different features (VGG, Glove; C3D, Glove; SF+C, C;) to organize experiments. This shows that the model always converges and achieves stable results for different initializations. This phenomenon demonstrates the ability of the model to learn to generate real spans from arbitrary random spans.

## C.3  Ablation study

**Different sampling strategies.** Denoising Diffusion Probabilistic Models (DDPM) [49] and Denoising Diffusion Implicit Models (DDIM) [69] are popular and classic diffusion models. We show the results of both strategies in Tab. 10(a). We find that DDIM and DDPM perform similarly, but DDIM samples faster. Therefore we adopt DDIM as the default technology.

**Effect of the schedule of $\beta$.** The schedule of $\beta$ determines the weighting ratio of different intensity noises. In Tab. 10(b), we find that the cosine schedule works better in our experiment. The cosine schedule makes the noisy spans change slowly at the beginning and end during the diffusion process, and the generation effect is more stable. So we set the cosine schedule as the default.

**Effect of the Box Renewal.** Box Renewal is a post-processing technique in DiffusionDet [61]. Tab. 10(c) shows that Box Renewal can indeed slightly improve the results. To keep the inference process as simple as possible, we do not use Box Renewal by default.

**Effect of the batch size.** As shown in Tab. 10(d), we set the batch size to 32 to achieve the best results.

**Effect of the layer number.** In Tab. 10(e), we show the effect of the layer number of Similarity-aware Condition Generator (SCG) and Video Moment Denoiser (VMD). The default setting (2+2:2): SCG contains 2 cross-attention and 2 self-attention layers and VMD contains 2 cross-attention layers. The default setting works best.

**Effect of the weight value on $\mathcal{L}_{sim}$.** We set the weight values of $\mathcal{L}_{sim}$ and $\mathcal{L}_{vmr}$ to 4 and 1, respectively. Keeping the weight value of $\mathcal{L}_{vmr}$ unchanged, we organize the weight influence experiment of $\mathcal{L}_{sim}$, as shown in Tab. 10(f). The default setting works best.

**Effect of the number of spans on R1@0.5.** Our training and inference are decoupled. Our simple framework allows us to input any number of random noises. Tab. 10(g) shows that there is a slight improvement when testing with more noise boxes.

Table 10: Ablation study (%) on the Charades-STA dataset with SF+C video features and CLIP text features. We report R1@0.5, R1@0.7 and $\text{MAP}_{avg}$. Default settings are marked in blue.

(a) Different sampling strategies.

| Sampling | R1@0.5 | R1@0.7 | $\text{MAP}_{avg}$ |
|---|---|---|---|
| DDPM [49] | **55.62** | 32.37 | 32.48 |
| DDIM [69] | 55.57 | **32.42** | **32.85** |

(b) Effect of the schedule of $\beta$.

| Schedule | R1@0.5 | R1@0.7 | $\text{MAP}_{avg}$ |
|---|---|---|---|
| Linear | 54.76 | 31.43 | 31.59 |
| Cosine | **55.57** | **32.42** | **32.85** |

(c) Effect of the Box Renewal [61].

| Schedule | R1@0.5 | R1@0.7 | $\text{MAP}_{avg}$ |
|---|---|---|---|
| w/o Box Renewal | 55.57 | 32.42 | 32.85 |
| w/ Box Renewal | **56.03** | **32.64** | **33.18** |

(d) Effect of the batch size.

| batch size | R1@0.5 | R1@0.7 | $\text{MAP}_{avg}$ |
|---|---|---|---|
| 16 | 52.42 | 30.81 | 30.14 |
| 32 | **55.57** | **32.42** | **32.85** |
| 64 | 53.78 | 32.25 | 31.93 |

(e) Effect of the layer number.

| SCG:VMD | R1@0.5 | R1@0.7 | $\text{MAP}_{avg}$ |
|---|---|---|---|
| 1+1:1 | 51.74 | 28.97 | 29.82 |
| 2+2:2 | **55.57** | **32.42** | **32.85** |
| 3+3:3 | 54.39 | 32.16 | 32.54 |

(f) Effect of the weight value on $\mathcal{L}_{sim}$.

| Weight | R1@0.5 | R1@0.7 | $\text{MAP}_{avg}$ |
|---|---|---|---|
| 1 | 52.34 | 29.63 | 29.98 |
| 2 | 53.74 | 30.82 | 31.15 |
| 4 | **55.57** | **32.42** | **32.85** |

(g) Effect of the number of spans on R1@0.5.

| train \ test | 1 | 3 | 5 | 10 | 20 |
|---|---|---|---|---|---|
| 1 | 50.83 | 50.91 | 50.97 | 50.87 | 50.82 |
| 3 | 53.98 | 54.12 | 54.19 | 54.23 | 54.21 |
| 5 | 55.36 | 55.41 | 55.57 | **55.69** | 55.51 |
| 10 | 53.71 | 53.84 | 53.86 | 53.89 | 53.93 |
| 20 | 53.16 | 53.38 | 53.40 | 53.44 | 53.42 |

# D  Broader Impacts

First, our work does not involve private data. Second, we believe that AI is a double-edged sword, and our model is no exception. For example, when users or websites use our model, only natural language is needed to locate video moments and collect desired video material, which improves the productivity of society. However, this may have a negative impact if the natural language entered by the user contains words related to violence, pornography, etc. We will consider these scenarios and implement a more secure VMR model.

