# OpenReview forum: "MomentDiff: Generative Video Moment Retrieval from Random to Real"
_NeurIPS.cc/2023/Conference — NeurIPS 2023 poster_

### Official Review · Reviewer_uAtb · 2023-06-19

**Soundness:** 4 excellent
**Presentation:** 3 good
**Contribution:** 3 good
**Rating:** 5
**Confidence:** 5

**Summary:**

This paper tackles the video moment retrieval task from the generative perspective and proposes a diffusion-based localization model, named MomentDiff. It could sample random temporal segments as initial guesses and iteratively refine them to generate an accurate temporal boundary. Moreover, this paper proposes two“anti-bias” datasets with location distribution shifts to evaluate the influence of location biases. Experiments on three public datasets validate the effectiveness of the proposed approach.

**Strengths:**

1. This paper addresses the cross-modal moment retrieval task using a diffusion-based model, which is interesting.
2. This paper builds two datasets with location distribution shifts, which is valuable for this research community.
3. Experiments on three datasets: Charades-STA, QVHighlights, and TACos, demonstrating the effectiveness of the proposed approach MomentDiff.


**Weaknesses:**

1. Despite the widespread use of datasets like TAcos, Charades-STA, and ActivityNet Captions, this paper chose not to conduct experiments using ActivityNet Captions.
2. Previous studies [1][2] employed CharadesCD and ActivityNet-CD to examine the influence of location biases. Nevertheless, this paper made the decision not to directly employ these datasets. Why?
[1] Towards Debiasing Temporal Sentence Grounding in Video
[2] A Closer Look at Temporal Sentence Grounding in Videos: Datasets and Metrics
3. To provide comprehensive evaluation, comparisons with other supervised, weakly supervised, and zero-shot moment retrieval methods are crucial. Examples of such methods include [3] DORi: Discovering Object Relationships for Moment Localization of a Natural Language Query in a Video, [4] Structured Multi-Level Interaction Network for Video Moment Localization via Language Query, [5] Multi-Modal Relational Graph for Cross-Modal Video Moment Retrieval, and [6] Language-free Training for Zero-shot Video Grounding.

**Questions:**

1. Could you please explain how to obtain the values of $Q_{\hat{v}}$, $K_{\hat{v}}$, and $V_{\hat{v}}$ mentioned on page 4, line 146?
2. Regarding the use of span embedding as the query in Intensity-aware attention instead of combining it with the textual query, could you please elaborate on the reasoning behind this decision? Additionally, it would be helpful to know if any experiments were conducted to validate this choice and provide justification.
3. This paper aims to incorporate audio features and integrate multi-modal video information. Could you please explain the methodology used to integrate the multi-modal video information? Furthermore, it is important to elaborate on how the paper demonstrates that the performance improvements are not solely a result of introducing audio information.

---

> ### Author Rebuttal · Authors · 2023-08-08
>
> **Q1 why not organize experiments on ActivityNet Captions?**
> We explore the VMR problem based on DETR. The datasets used by the representative methods (MomentDETR and UMT) are QVHighlights and Charades-STA, so we use the same datasets.
> Besides, the results of our method on ActivityNet Captions (C3D features) are shown below, where we record the 1 epoch training time and the inference time for all test samples in one A100 GPU:
> |Method|R1@0.3|R1@0.5|R1@0.7|${MAP}_{avg}$|Training time|Testing time|
> |:---|---:|---:|---:|---:|---:|---:|
> |2DTAN [15]|59.92|44.63| 27.53|27.26|1.1h|523.74s|
> |MMN [16]|65.21|48.26|28.95|28.74|1.5h|662.12s|
> |MomentDETR [25]|61.87|43.19|25.74| 25.63|0.05h|52.72s|
> |**Ours**|62.79|46.52|28.43|28.19|0.06h|91.43s|
>
> 1. Our model has been improved compared with baseline (MomentDETR). Compared to SOTAs, we still have competitive results.
> 2. Compared with MMN, our method has **25 times** faster training time per epoch and **7.24 times** faster testing time.
>
> We will add more results in the revised version.
>
> **Q2 why not choose Charades-CD and ActivityNet-CD to organize experiments?**
> Thanks for your constructive suggestion.
> In VMR task, the span center $c\_{0}$ and length $w\_{0}$ are two important parameters.
> So we want to verify the model generalization from two perspectives of length and position (Charades-STA-Len and Charades-STA-Mom).
> The dataset construction strategy is very simple, and the evaluation perspective is comprehensive.
>
> As suggested, we use the same VGG features to organize OOD experiments on **Charades-CD**:
> |Method|R1@0.3|R1@0.5|R1@0.7|$MAP_{avg}$|
> |:---|---:|---:|---:|---:|
> |2DTAN [15]|49.71|28.95|12.78|12.60|
> |MMN [16]|55.91|34.56|15.84|15.73|
> |MomentDETR [25]| 57.34|41.18|19.31|18.95|
> |**Ours**|67.73|47.17|22.98| 22.76 |
>
> In the **ActivityNet-CD** dataset, we use the same C3D feature to organize OOD experiments:
> |Method|R1@0.3|R1@0.5|R1@0.7|$MAP_{avg}$|
> |:---|---:|---:|---:|---:|
> |2DTAN [15]|40.04|22.07|10.29|12.77|
> |MMN [16]|44.13|24.69|12.22|15.06|
> |MomentDETR [25]|39.98|21.30|10.58|12.19|
> |**Ours**|45.54|26.96|13.69|16.38|
>
>
> Conclusion:
> 1. **On Charades-CD and ActivityNet-CD, we exceed baseline (MomentDETR) by a large margin.**
> Although MMN achieved SOTA results on ActivityNet in the answer for Q1,  MMN (R1@0.5: 24.69) is still lower than our model (R1@0.5: 26.96) on ActivityNet-CD.
> 2. These results prove the robustness of the model in dealing with OOD scenarios.
> Our generative framework alleviates the location biases problem.
> We will add **the above results and codes** in the revised version, which make our paper more convincing.
>
> **Q3 comparisons with other supervised, weakly supervised, and zero-shot methods.**
> Very good suggestion.
> We will refer to and compare these works in the revised version.
> For the fairness of the experiment, we use the same features for comparison.
>
> Charades-STA:
> |Method|Type|R1@0.5|R1@0.7|
> |:---|---:|---:|---:|
> | DORi[3] | VGG |  43.47  |  26.37  |
> | Ours |VGG| 51.94  | 28.25  |
> | SMIN[4] |C3D|   50.32 |  28.95  |
> | MMRG [5] |C3D|  44.25 | - |
> | Ours |C3D|  53.79 | 30.18 |
>
> ActivityNet:
> |Method|Type|R1@0.5|R1@0.7|
> |:---|---:|---:|---:|
> | ZSVG[6] |C3D|   32.59 |  15.42  |
> | Ours |C3D|  46.52 |28.43 |
>
> **Q4 explain how to obtain $Q_{\hat{v}}$, $K_{\hat{v}}$ and $V_{\hat{v}}.$**
> Sorry for the confusion.
> We get $Q_{\hat{v}}$, $K_{\hat{v}}$ and $V_{\hat{v}}$ through three different linear projection:
> $Q_{\hat{v}} = W_{q}\hat{V}, K_{\hat{v}} = W_{k}\hat{V}, V_{\hat{v}} = W_{v}\hat{V}, $
> where $W_{q}$, $ W_{k}$ and $W_{v}$ are linear projection matrices, $\hat{V}$ is the embedding output by the previous cross-attention layers.
> We will describe this process clearly in the revised version.
>
> **Q5 the reason for using span embeddings as queries instead of combining it with the text.**
> There are three reasons:
> 1. The text has already interacted with the video in the Similarity-Aware Condition Generator, so continuing to add text information may be unnecessary.
> 2. Usually the query of the decoder in DETR series work (such as MomentDETR) is learnable embeddings, not text features.
> Learnable queries introduce location bias inherent in the dataset, so we set the query to be data-independent random spans.
> Besides, our model is based on the diffusion model. The entire training process is the noise addition and denoising process of ground truth spans. Both the input query and output results should be consistent spans.
> 3. We add text features to the query and find that the results on Charades-CD is reduced. We speculate that adding text features to the query may perturb the denoiser to perceive the noise intensity, and thus the results will decrease. The results on Charades-CD are as follows:
> |Method|R1@0.3|R1@0.5|R1@0.7|$MAP_{avg}$|
> |:---|---:|---:|---:|---:|
> |w/ text features|67.21|45.99|22.63|22.26|
> |Ours|67.73|47.17|22.98|22.76|
>
> **Q6 explain the method used to integrate multi-modal video information.**
> The key to this paper is how to alleviate the location bias problem and iteratively generate accurate temporal spans.
> For adding additional audio information, our purpose is only to prove that our method can expand more modalities, but this is not the focus of the paper.
>
> In addition, the process of fusing audio information is very simple, we only concatenate the audio features and the input visual features (e.g., VGG features).
> As shown in Table 1 of our main paper, the results are as follows:
>
> |Method|Type|R1@0.5|R1@0.7|$MAP_{avg}$|
> |:---|---:|---:|---:|---:|
> |UMT [26]|VGG+Audio|48.44|29.76|30.37|
> |MomentDiff|VGG|51.94|28.25|31.66|
> |MomentDiff|VGG+Audio|52.62|29.93|31.81|
>
> From the results, we can find:
> 1. Without using audio, the model still can achieve good results, such as R1@0.5=51.94.
> 2. Using the same settings (VGG+Audio), our model (R1@0.5=52.62) can exceed UMT (R1@0.5=48.44) by a large margin.
>
> The above results and more results in the paper prove the effectiveness of the model itself.

---

> > ### Comment · Reviewer_uAtb · 2023-08-13
> > **Response to authors**
> >
> > The authors' response addresses some of my concerns. I will adjust my score to "borderline accept."

---

### Official Review · Reviewer_mu2A · 2023-06-28

**Soundness:** 3 good
**Presentation:** 4 excellent
**Contribution:** 4 excellent
**Rating:** 7
**Confidence:** 4

**Summary:**

To deal with the problem of temporal location bias, the authors propose a diffusion-based video moment retrieval framework, MomentDiff. They introduce the diffusion process into temporal localization from a generative perspective, and gradually generate real span coordinates from coarse to fine. Compared to learnable queries, the random noise input to the model reduces the dependence on the location information of the dataset. Therefore, MomentDiff achieves better results on two "anti-bias" datasets with changing location distributions. Besides, MomentDiff consistently outperforms state-of-the-art methods on three public benchmarks.

**Strengths:**

a) This work proposes a novel and effective diffusion framework on the video moment retrieval task and alleviates the important location bias problem. The paper is also well motivated and well written.
b) To demonstrate the robustness of the model, they propose two anti-bias datasets, which seem to be one of the main contributions of the paper.  Promising experimental results.
c) The paper presents promising experimental results. Authors will provide code and datasets.

**Weaknesses:**

While I don't see obvious weaknesses, there are a few minor suggestions, and additional questions that the author needs to answer carefully:
a) Please revise the notation and typos of the paper. For example, \epsilon in Eq 6 is not clearly defined above, only \epsilon_m.
b) Please re-check the paper and correct errors on formatting, grammar, etc. For example, L154. The Eq (1) uses “Snj”, “Spj”, but L154 writes “Spj”, “Sni”.
c) Some of the figures in Fig. 1 and Fig. 3 are so small that they are difficult to see even when zoomed in.

**Questions:**

In image generation tasks, images generated by diffusion models are often diverse. With different random noise inputs, are the span coordinates generated by the MomentDiff model for the same video-text pair quite different, and is the model performance stable? If the result is relatively stable, what is the reason?

**Limitations:**

The paper argues that if the user enters words that violate the law, the model may have a potential negative impact. I think sensitive word filtering technology can effectively solve this problem.

---

> ### Author Rebuttal · Authors · 2023-08-08
>
> Thank you very much for your appreciation of our paper, including the motivation, writing, and robustness of the model. We will carefully revise the paper according to the questions and suggestions raised by the reviewers.
>
> **Q1 typos and errors in formatting, grammar, and figures.**
>
> Thank you so much for your constructive comments and review.
> We will carefully fix these errors, including formulas, font sizes, and spelling mistakes.
>
> **Q2 is the model performance stable?**
>
> In Figure 1 in our Supplementary Material, we show the performance of the model with multiple random seeds (seed= 2023, 2022, 2021, 2020, 2019) on the Charades-STA dataset.
> For readability, we present it here in tabular form:
>
> |Type|R1@0.5|R1@0.7|MAP@0.5|MAP@0.75|$MAP_{avg}$|
> |:---|---:|---:|---:|---:|---:|
> |VGG, Glove|51.94 $\pm$ 1.9|28.25 $\pm$ 1.7|59.86 $\pm$ 2.4|29.11 $\pm$ 0.6|31.66 $\pm$ 0.4|
> |C3D, Glove|53.79 $\pm$ 2.1|30.18 $\pm$ 0.7|59.32 $\pm$ 1.6|29.85 $\pm$ 0.4|31.89 $\pm$ 0.4|
> |SF+C, C|55.57 $\pm$ 0.9|32.42 $\pm$ 1.8|61.07 $\pm$ 2.6|32.51 $\pm$ 1.5|32.85 $\pm$ 0.9|
>
>
> This shows that the model always converges and achieves stable results for different initializations.
>
> We believe that the reason is related to the characteristics of the diffusion model itself and the loss constraints.
> Specifically, during the training process of the model, the input is the noisy span, and we constrain the model to generate a real span.
> The real span has fixed manual time annotation, which is obviously different from the image generation task.
> So the model will not generate overly diverse results under the constraints of annotations and losses.
>
> In addition, since the input noisy spans are random, the model needs to learn the ability to generate real spans from arbitrary spans.
> This ability ensures that the model can perform well under different initializations to a certain extent.
>
> **Q3 recommendations to address potential negative impacts.**
>
> Thank you for your very professional review. We will investigate recent projects on sensitive word filtering and add to the supplementary material.

---

> > ### Comment · Reviewer_mu2A · 2023-08-17
> >
> > Thanks for your answer. After reading the author's comments as well as other reviewers' concerns, most of my concerns were resolved, so I tend to keep the paper score unchanged. Also, I agree with reviewer uAtb that it is important to add the experimental results of ActivityNet, CharadesCD and ActivityNet-CD datasets, which will make the analysis of the paper more comprehensive.

---

### Official Review · Reviewer_2CFW · 2023-07-04

**Soundness:** 3 good
**Presentation:** 3 good
**Contribution:** 3 good
**Rating:** 7
**Confidence:** 5

**Summary:**

This paper first tackles video moment retrieval from a generative perspective, and proposes a novel framework called MomentDiff based on recently proposed technique Diffusion Models. MomentDiff can generate correct results from random spans, which can resist the temporal location biases. The experiments on three public datasets and two anti-bias datasets proposed by the authors demonstrate the effectiveness of MomentDiff.

**Strengths:**

1.	The generative perspective for video moment retrieval is novel.
2.	The designed MomentDiff is effective for the location bias problem and easy to reproduce.
3.	Experimental results on three public datasets and two anti-bias datasets demonstrate the effectiveness of the proposed method.


**Weaknesses:**

1.	From a generative perspective, traditional generative models like GANs can also be applied to the video moment retrieval task. Do the authors believe that these methods can be used, and if so, what is the difference between GAN and Diffusion models in this task? If not, please provide a reason.
2.	It is unclear how other methods have solved the problem of location biases. It would be helpful for the authors to compare and contrast the advantages of the proposed method with existing solutions.
3.	The total loss function is missing. Please clarify whether both loss functions L_{sim} and L_{vmr} are weighted 1.


**Questions:**

See the Weaknesses.

**Limitations:**

The author points out that too many iterations will slow down the inference speed, and the solution is to reduce the number of iterations and make a trade-off between performance and speed. It is recommended that the author give the iteration round parameters on all datasets to show better trade-off.

---

> ### Author Rebuttal · Authors · 2023-08-08
>
> Thanks for your constructive comment. We answer your questions point by point.
>
> **Q1 Using GAN in VMR.**
> We tend to use diffusion models instead of GANs to build generative VMR frameworks for three reasons:
> 1. The VMR task requires the model to locate fine-grained moments, so a single-step generative model like GAN may not work well.
> In our experiments, the effect of MomentDiff at step=1 is significantly lower than that of subsequent iterative generation results.
> 2. In addition, the diffusion model is very flexible and easy to calculate, we can choose any number of spans and steps to solve the VMR task.
> 3. The training process of GAN is difficult and unstable.
>
> **Q2 existing methods about location bias.**
> To remove the harmful location bias, DCM [1] first disentangles the moment representation and applies causal intervention on the multimodal model input to remove the confounding effects of moment location.
> [2] samples from the training set and constructs the uniform dataset, and reduces the gradient of biased samples to achieve an unbiased model.
> We propose a new diffusion VMR scheme from a generative perspective, which mitigates location bias by replacing learnable queries with random noise.
> We believe that generative methods will bring new insights to the field.
> We will carefully emphasize these methods in related work.
>
> [1] Deconfounded Video Moment Retrieval with Causal Intervention --SIGIR 2021
>
> [2] Towards Debiasing Temporal Sentence Grounding in Video --arXiv:2111.04321
>
> **Q3 Weight value on $L_{sim}$ and $L_{vmr}$.**
> We set the weights of $L_{vmr}$ and $L_{sim}$ to 1 and 4, respectively. Keeping the weight of $L_{vmr}$ unchanged, we organize the weight influence experiment of $L_{sim}$ on Charades-STA (VGG features), as shown in the following table:
> | $\lambda_{L_{sim}}$ | R1@0.5 | R1@0.7 | MAP@0.5 | MAP@0.75| $\text{MAP}_{avg}$ |
> |:--- | :----: | ---:|  ---:|  ---:|  ---:|
> |1|  50.21 | 27.42  | 58.33 | 28.02 | 30.17|
> |2|  51.23 | 27.95  | 59.55 | 28.83 | 31.19|
> |4| **51.94**  | **28.25**  | **59.86** | **29.11** | **31.66**|
> |8|  51.39 | 28.01  | 59.42 | 28.89 | 30.92|
>
> More results will be added to the supplementary material.
>
>
> **Q4 the iteration round parameters on all datasets.**
> We show the R1@0.7  results corresponding to different iteration steps (1, 2, 10, 50, 100) on the three datasets in the table below:
>
> | Dataset | Type | 1 | 2 | 10 | 50| 100|
> |:--- | :----: | ---:|  ---:|  ---:|  ---:|     ---:|
> | Charades-STA |VGG,Glove | 26.12  |  27.93 | 28.21| **28.25**| 28.27|
> | QVHighlights | SF+C, C  |  37.47  |  39.42 | 39.59| **39.66**| 39.58|
> | TACoS | C3D,Glove  |  15.24  |  16.81 | 17.69| **17.83**| 17.97|
>
> More results will be added to the supplementary material.

---

> > ### Comment · Reviewer_2CFW · 2023-08-21
> >
> > Thanks, the responses effectively address my concerns.
> >
> > I do not have further questions. This paper is the first to propose a generative algorithm framework in VMR, and alleviates the current important location bias problem. After reading all reviewer responses, I think the algorithm performs well in OOD scenarios and outperforms existing methods. Therefore, I recommend accepting this paper.

---

### Official Review · Reviewer_KdBK · 2023-07-04

**Soundness:** 3 good
**Presentation:** 3 good
**Contribution:** 3 good
**Rating:** 6
**Confidence:** 3

**Summary:**

This paper proposes a novel generative approach, MomentDiff, to address the Video Moment Retrieval (VMR) task. It replaces traditional dense or learnable proposals with random spans and a diffusion-based denoiser to refine predictions, mimicking the human process of identifying key video moments. This reduces the impact of temporal location biases and improves the system's generalizability. The authors also introduce two "anti-bias" datasets, Charades-STA-Len and Charades-STA-Mom, for evaluation. Experiment results showed that MomentDiff outperforms existing methods in efficiency and transferability.

**Strengths:**

1. The proposed method creatively combines pre-trained video and text backbones for feature extraction, a similarity-aware condition generator, and a video moment denoiser. This composite approach takes existing tools and blends them in a unique way. The inclusion of audio data as a feature, alongside visual and textual data, also represents an innovative approach to video moment retrieval.
2. The paper showcases a high-quality approach by incorporating various feature extractors, utilizing a multilayer transformer for multimodal interaction, and deploying a similarity-aware fusion embedding. The fact that the paper also discusses the limitations of the proposed method speaks to its quality and rigor.
3. The proposed methodology is outlined clearly and in a structured manner. Each part of the system, from feature extraction to the inference process, is explained with sufficient detail. However, some areas could benefit from additional explanation (e.g., the impact of the quality of fusion embeddings on the denoising process), which could further enhance clarity.
4. The paper tackles the important problem of video moment retrieval, which has broad implications in fields like media indexing, recommendation systems, and video summarization. The solution proposed in the paper, especially with the inclusion of audio features, can be significant in improving the efficiency and effectiveness of video moment retrieval tasks. By outlining its method clearly and discussing potential limitations, the paper contributes to further research and improvement in the field.

**Weaknesses:**

1. The proposed method relies heavily on the effectiveness of the chosen feature extractors. Although they have tested multiple feature extraction models, the paper does not discuss the impact of these choices on the final results in detail. Additionally, the models chosen for feature extraction could potentially limit the generalizability of the approach to datasets significantly different from those on which the models were trained.
2. The paper does not provide a clear comparison with existing methods in terms of computational resources. This makes it hard to gauge the improvement the proposed method offers over current techniques.
3. The paper mentions multiple hyperparameters but does not discuss how they are selected or tuned. This could impact the replicability and robustness of the model across different datasets.


**Questions:**

1. Clarification on Visual and Textual Representations: It would be helpful if the authors could elaborate on why they chose the specific visual and textual extractors, like VGG, C3D, CLIP, Glove, etc. Are there specific reasons these were chosen over other potential extractors?
2. Elaboration on Span Generation Process: In the span generation process, it is mentioned that for the same video, the correct video segments corresponding to different text queries are very different. Could you elaborate more on this? Is there a way to address this challenge?
3. Justification for Hyperparameter Choices: Could the authors provide further clarification on the selection of the hyperparameters used in the model? How were these optimized, and what was the impact on model performance?
4. Scalability of the Model: Could the authors discuss how this model scales with larger, more complex datasets? Can the method efficiently handle real-world scenarios with high volumes of data, and if so, are there any limitations or performance degradation?
5. Use of Pre-Trained Models: What are the implications of using several pre-trained models? How does it affect the generalizability of the proposed method across diverse datasets, especially ones that differ significantly from the datasets these pre-trained models were trained on?
6. Computational Resources: Could the authors provide details about the computational resources required for the model to run both in the training and inference stages? This is crucial for evaluating the practicality of the proposed model.
7. No motivation is provided in Similarity-aware Condition Generator, i.e., why specific modality features are selected as Query, Key, and Values? Why not any other combination?


**Limitations:**

1. Authors have provided limited limitations.
2. Code is not provided in supplementary that can help with in more detailed understanding.

---

> ### Author Rebuttal · Authors · 2023-08-08
>
> **Q1 the reason and impact of extractors.**
> Thank you for suggestions.
> We chose these extractors for two reasons:
> 1. We want to prove that our method is general to different types of extractors, so different models are used: 1)2D encoder, VGG. 2)3D encoder, C3D. 3)Cross-modal encoder, CLIP.
> Many methods only use a single extractor.
> 2. The extractors involved in the VMR task include VGG, C3D, CLIP. For the fairness of the comparison, we use the consistent pre-trained models.
>
> The impact of different feature extractors:
>
> We agree with your profound point: different pretrained models generalize differently on downstream tasks.
> VGG, C3D or CLIP are pre-trained on ImageNet, Sports-1M, or 4B image-text pairs respectively.
> However, since our video data is quite different from the above datasets, the impact of pre-training model may not lie in whether the data has been seen, but in the representation ability of pre-training model itself.
> VGG is an early image backbone, and its representation ability in VMR may be weaker than that of C3D and CLIP.
> Therefore, on Charades-STA, VGG is not as effective as C3D or CLIP.
>
> We will emphasize the impact of model choices in the revised version.
>
> **Q2 computing resources in training and testing.**
> We count the training time (Tr) of an epoch and the inference time (In) on Charades-STA with VGG features.
>
> Charades-STA:
> |Method|R1@0.5|R1@0.7|Tr|In|
> |:---|---:|---:|---:|---:|
> |MMN [16]|46.93|27.07|479.63s|53.42s|
> |MomentDETR [25]|50.54|28.01|40.74s|12.42s|
> |Ours-step 1|49.17|26.39|48.12s|7.56s|
> |Ours-step 2|50.81|27.84|48.12s|8.23s|
> |Ours-step 10|52.36|28.08|48.12s|11.01s|
> |Ours-step 50|51.94|28.25|48.12s|20.74s|
>
> 1. Compared with MMN, MomentDiff (Step=2) achieves better results and improves training speed by 10 times and inference speed by 7 times.
> Because MMN predefines lots of proposals, which increase computational overhead.
> 2. All experiments are performed on one A100 GPU.
> Memory usage is correlated with number and dimension of features .
> For VGG, C3D or SF+C, we extract video features every 1/6s, 1s or 1s on charades-STA.
> We use VGG (4096-d), C3D (500-d), or SF+C (2816-d) features with 40.1G, 3.86G, or 4.02G memory.
> Even using C3D features, MomentDiff outperforms SOTA models.
> We can flexibly select extractors according to our own resources.
>
> **Q3 the impact of hyperparameters.**
> Thank you for good suggestion.
> First, we provide the ablation study about scale factor, span number and batch size in Table 5 of our main paper and Table 3 of supplementary material.
>
> Next, we show more results on Charades-STA (SF+C features), including the loss weight $\lambda_{L1},\lambda_{iou},\lambda_{ce}$ and the weights $\lambda_{L_{sim}}$ of $L_{sim}$, Transformer layer number.
>
> 1. We show the results of different weights and $[\lambda_{L1}=10,\lambda_{iou}=1,\lambda_{ce}=4, \lambda_{L_{sim}}=4]$ is best.
>
> | $\lambda_{L1}$ | $\lambda_{iou}$ | $\lambda_{ce}$ |  $\lambda_{L_{sim}}$ |R1@0.5 | R1@0.7| $MAP_{avg}$|
> |:---|---:|---:|---:|---:|---:|---:|
> |10|1|4|4|55.57|32.42|32.85
> |10|1|4|2|55.03|31.94|32.22
> |10|1|4|1|54.36|31.14|31.09
> |5|1|4|4| 53.74|30.25| 30.48|
> |10|2|4|4|55.42|32.29|32.57|
>
> 2. We show the effect of the layer number of Similarity-aware Condition Generator (SCG) and Video Moment Denoiser (VMD).
> The default setting: SCG contains 2 cross-attention and 2 self-attention layers and VMD contains 2 cross-attention layers.
> The default setting works best:
> |SCG|VMR|R1@0.5|R1@0.7|$MAP_{avg}$|
> |:---|---:|---:|---:|---:|
> |1+1|1|51.74|28.97|29.82|
> |2+2|2|55.57|32.42|32.85|
> |3+3|3|54.39|32.16|32.54|
>
> **Q4 elaboration on Span Generation Process.**
> In datasets, there are multiple segment-text pairs for the same video.
> The content of text is different, it will correspond to different segments.
> We designed two solutions to this issue:
> 1. We use cross-attention layers in SCG. The video and text in the cross-attention layers are fully interacted, so that the model can perceive different video-text pairs.
> 2. We design $\lambda_{L_{sim}}$ to optimize the fusion embedding $F$.
> Note that $F \in \mathbb{R}^{N_v * d}$, where N_v is the number of frames.
> The purpose of loss is to pay attention to the frames within ground truth spans.
> In this way, for different segment-text pairs of the same video, the model can focus on positive frames to generate accurate spans.
>
> We will add explanations.
>
> **Q5 using complex datasets.**
> Thanks for valuable question. We organize experiments on ActivityNet Captions, which contains complex activities.
> There are 37417, 17505, and 17031 pairs for training, validation, and testing. The results on ActivityNet Captions (C3D features):
> |Method|R1@0.3|R1@0.5|R1@0.7|$MAP_{avg}$|
> |:---|---:|---:|---:|---:|
> |MomentDETR|61.87|43.19|25.74|25.63|
> |Ours|62.79|46.52|28.43|28.19|
>
> Compared with baseline, our method still has a certain improvement.
>
> To verify performance in real world, we refer to the Out of Distribution (OOD) division on ActivityNet-CD[1]. The results are:
> |Method|R1@0.3|R1@0.5|R1@0.7|$MAP_{avg}$|
> |:---|---:|---:|---:|---:|
> |MomentDETR|39.98|21.30|10.58|12.19|
> |Ours|45.54|26.96|13.69|16.38|
>
> The results prove model robustness.
> We will add results in the revised version.
>
> [1] A Closer Look at Temporal Sentence Grounding in Videos: Dataset and Metric.
>
> **Q6 why specific modality features are selected as Q, K, V?**
> Because the task purpose is to locate the start and end positions.
> Therefore, we hope that the features after multi-modal fusion are frame-level features, e.g., $ F \in \mathbb{R}^{N_v * d}$, which is convenient for frame-level loss design based on ground truth span.
>
> According to the attention formula in Transformer, the dimension of output embedding is consistent with the dimension of Query.
> Therefore, we need to design video features $ V \in \mathbb{R}^{N\_v * d} $ as Query and text features $ T \in \mathbb{R}^{N_t*d}$ as Key and Value.
> If $T$ is Query, $F$ is word-level, which may not be suitable for video to locate spans.

---

> ### Comment · Reviewer_KdBK · 2023-08-17
>
> After reviewing the authors' responses and considering feedback from other reviewers, I have decided to maintain my initial score.

---

### Official Review · Reviewer_oTs9 · 2023-07-05

**Soundness:** 2 fair
**Presentation:** 3 good
**Contribution:** 3 good
**Rating:** 5
**Confidence:** 4

**Summary:**

This paper first proposes a diffusion model for video moment retrieval to overcome proposal-based moment retrieval and distribution-specific methods.

**Strengths:**

[+] First work bridging generative frameworks into deterministic task as video moment retrieval.
[+] Illustrative presentation
[+] writing is easy to follow

**Weaknesses:**

[Motivation]
[1] There are many solutions (i.e. 2DTAN, VSLNet) without relying on moment proposals. Fully-supervised VMR does not concern about proposal generation, as it is frame-level supervision is available, where the previous works already design the regression-based (i.e., Attention Based Localization Regression) methods.

[2] As this paper proposes a new framework (diffusion framework) for the VMR, what the previous frameworks of VMR are relying on the distribution-specific proposals? This paper refers to the 2DTAN, the framework of 2D map with respect to start-time and end-time can allow all the possible moments, where the framework is not relying on the distribution-specific bias. I think the authors may understand the applying the masking in the map as resulting distribution-specific proposals, which is more related to the heuristic filtering by empirical experiments, not a bias problem.

[3] Why the diffusion (generative framework) more can be better than the deterministic models? In fact, the authors' proposed method is presented to overcome the proposal-learnable methods, however, current VMR method is not relying on proposals. (rather, the weakly-supervised method is dependent on proposals). Furthermore, location bias problem is sourced from dataset distributions, proposed diffusion framework does not correlate to mitigating bias.

[Method]

[1] Preliminary section is recommended in the paper or appendix about the forward-backward process of diffusion framework (e.g., denoising diffusion probabilistic model, conditional diffusion, sampling) to enhance the readability.

[2] x_{0} is a 2-dimensional vector of center point and width. Does this paper truly add Gaussian noise on to the 2 values and denoise them?

[Experiment]

[1] I can not trust the performances in Table. Is there any experimental qualitative evidence why denoising frameworks can guarantee more performance than previous work?


**Questions:**

I want to get answers about the weaknesses above in the rebuttal.

**Limitations:**

See in the weakness.

---

> ### Author Rebuttal · Authors · 2023-08-09
>
> Thank you for your comments! We hope our answers can address you concerns.
>
> [Motivation]
>
> Thank you for your very professional review.
> In fact, our understanding is basically the same, but there are two points that need to be clarified before solving your doubts.
>
> **Definition of proposal.** There are indeed many methods that do not require proposal generation. However, the proposal we refer to is a broad concept, including dense anchors in 2DTAN and learnable queries in MomentDETR. Like [1], we regard both 2DTAN and MomentDETR as  implicit proposal-based methods.
>
> [1] CONE: An Efficient COarse-to-fiNE Alignment Framework for Long Video Temporal Grounding
>
> **Promising DETR-based VMR framework.** Recent VMR methods are mainly divided into three types: dense proposal-based methods (e.g., 2DTAN, MMN), regression-based methods (e.g., ABLR, VSLNet) and DETR-based methods (e.g., MomentDETR, UMT). The disadvantage of dense proposal-based methods is that there is a large amount of proposal redundancy and the calculation speed is slow (refer to Q1's reply in Reviewer uAtb). Regression-based methods can alleviate this problem, but the results are often not good enough. Recently, DETR-based methods have weighed efficiency and performance. MomentDETR exceeds MMN 7 times in terms of inference speed (see Table 2 in Supplementary Material), and it also exceeds many regression-based methods in terms of performance.
> For example, MomentDETR can achieve R1@0.5 50.49 in Charades-STA, surpassing VSLNet(48.67) [18].
> Therefore, we further explore VMR tasks based on DETR, which is a very promising genre.
>
> **Q1 many proposal-free methods.**
> The emerging DETR-based VMR method is an important genre, and we are solving the location bias problem existing in the DETR series.
>
> **Q2 motivation about 2DTAN.**
> Sorry for the misunderstanding.
> We do not think that 2DTAN suffers from severe location bias problems.
> We introduced 2DTAN mainly to illustrate the shortcomings of dense proposals:
> these methods have a large redundancy of proposals and the numbers of positive and negative proposals are unbalanced.
>
> Besides, in Table 2 in the supplementary material, we also show the inference efficiency comparison between 2DTAN (42.18s) and our method (20.74s).
> This proves that dense proposals do affect model efficiency.
>
> **Q3.1 & Q6 why can the generative framework reach SOTA results? Qualitative analysis?**
> We improved based on MomentDETR. Both MomentDETR and UMT have SOTA performance, refer to Table 1 in our manuscript.
> Specific model differences include iterative denoising paradigm and training details.
> 1. Iterative denoising paradigm. In Table 5(c) w/o VMD, we find that removing the denoiser for model optimization, the final result drops by 8.88% in R1@0.5. According to Table 5(f), the result (53.31) of only 1-step denoising is about 2% lower than the best result (55.62).
> When the model performs 1-step denoising, it is actually similar to MomentDETR.
> The above results prove the effectiveness of the iterative denoising paradigm.
>
>     **Qualitative analysis.** In Figure 5 of our manuscript, we visualize the prediction results of MomentDETR and MomentDiff. **In Figure 1 of the rebuttal pdf file, we further present the results generated by our method in a single step.** The result of single-step generation still has a certain deviation from the correct video segments, especially the example on the right of Figure 1. Through continuous iterative diffusion denoising, our model can finally obtain better results than MomentDETR.
>
>
> 2. Encoder and loss functions.
> Compared with MomentDETR, we use additional cross-attention layers as encoder, which strengthens modality information interaction. We use point cross-entropy loss, which focuses on finer-grained and comprehensive positive and negative video frames.
>
> Definitely, we will present the above results, analysis and codes in the revised version.
>
>
> **Q3.2 reason about alleviating bias:**
> A simple idea is to solve it at the data level, which can be sampled from biased data sets or constructed with more evenly distributed data for training.
> But we hope to solve the problem from a deeper and essential perspective, that is, the model itself.
> Specifically, learnable queries in MomentDETR may tend to focus on video segments where locations in the dataset occur more often.
> Therefore, we directly replace the queries with data-independent random noise, which can alleviate the above-mentioned bias to a certain extent.
> In addition, more experimental results can refer to the reply of reviewer uAtb Q2. We achieve impressive performance on the Out of Distribution (OOD) evaluation dataset (Charades-CD and ActivityNet-CD).
>
>
>
> [Method]
>
> **Q4 preliminary section about diffusion process.**
> Thanks for your constructive suggestions.
> In the supplementary material, we have given the pseudo codes of diffusion training and inference in Algorithm 1 and 2 for the convenience of readers.
> To make it clearer, we will add the basic background, flowchart and more details of common diffusion models in detail in the appendix.
>
> **Q5 add noise to the span.**
> Yes，we add Gaussian noise according to the following formula:
> $\boldsymbol{x}_m=\sqrt{\bar{\alpha}_m} \boldsymbol{x}_0+\sqrt{1-\bar{\alpha}_m} \boldsymbol{\epsilon}_m$ and code:
> ```python
>  #python3
> noise = torch.randn(self.query_num, 2).cuda()
> sqrt_alphas_cumprod_m = extract(self.sqrt_alphas_cumprod, m, x_0.shape)
> sqrt_one_minus_alphas_cumprod_m = extract(self.sqrt_one_minus_alphas_cumprod, m, x_0.shape)
> x_m =  sqrt_alphas_cumprod_m * x_0 + sqrt_one_minus_alphas_cumprod_m * noise
> ```
> In addition, the denoising process is inspired by the diffusion model that can exploit random noise to generate images in specified semantics.
> We generate noise into the temporal spans corresponding to the query semantics, which can be fully realized through the guidance of video-text fusion embeddings and appropriate loss constraints.
> We will publish code and models.

---

> > ### Comment · Reviewer_oTs9 · 2023-08-18
> >
> > My questions are resolved well. It is highly recommended to release the code publicly available for enhancing reproducibility. I raise my score. Thank you!

---

> ### Comment · Area_Chair_YV6q · 2023-08-17
> **Request for your feedback in light of authors' feedback**
>
> Thank you for your valuable insights and expertise which have contributed significantly to the review process.
>
> Following the initial review, the authors have provided a detailed rebuttal addressing the feedback and comments provided by our esteemed reviewers, including yourself. I kindly request that you take the time to carefully review the authors' rebuttal and assess its impact on your initial evaluation.
>
> Please share your thoughts and any additional points you may have after reading the authors' rebuttal. Thank you very much!

---

### Author Rebuttal · Authors · 2023-08-09

Thanks to all reviewers for their careful comments.

We would like to thank all reviewers for their appreciation of our paper, including **"writing is easy to follow" (Reviewer oTs9), "the high-quality approach" (Reviewer KdBK), "a novel framework" (Reviewer 2CFW), "promising results" (Reviewer mu2A) and "valuable datasets" (Reviewer uAtb).**
Our contributions specifically come from three aspects:

1. we are the first to tackle video moment retrieval from a generative perspective, which  mitigates temporal location biases from datasets.
2. We propose a new framework, MomentDiff, which utilizes diffusion models to iteratively denoise random spans to the correct results.
3. We propose two “anti-bias” datasets with location distribution shifts to evaluate the influence of location biases. Extensive experiments demonstrate that MomentDiff is more efficient and transferable than state-of-the-art methods on three public datasets and two anti-bias datasets.

In addition, we mainly reply to the reviewers from three aspects:
1. Clarify the motivation and the reasons for the module design.
2. We prove the effectiveness of the model in a larger dataset and more OOD datasets.
3. We show more ablation studies.

**Please refer to individual responses to each reviewer for specific details.**

Finally，**thanks again to the reviewers for their efforts, we have benefited a lot.**

---

### Decision · Program_Chairs · 2023-09-21

**Decision:**

Accept (poster)

**Comment:**

This paper receives final review scores of 7/5/5/7/6.

The paper presents the first attempt to tackle video moment retrieval problem from a generative perspective. It also presents two anti-bias datasets for evaluating the influence of location bias. The reviewers appreciate the novelty of the work and the illustrative presentation which is easy to follow. The experimental results are also comprehensive and convincing.

All reviewers acknowledged that the responses effectively addressed their concerns, so they either kept their original rating or raised the rating.